# Acidity as Descriptor for Methanol Desorption in B-, Ga- and Ti-MFI Zeotypes

**Simone Creci** [1,*] [iD], **Anna Martinelli** [2] [iD], **Szilvia Vavra** [2] [iD], **Per-Anders Carlsson** [1] [iD] and **Magnus Skoglundh** [1] [iD]

[1] Competence Centre for Catalysis, Department of Chemistry and Chemical Engineering, Chalmers University of Technology, 41296 Gothenburg, Sweden; per-anders.carlsson@chalmers.se (P.-A.C.); skoglund@chalmers.se (M.S.)

[2] Department of Chemistry and Chemical Engineering, Chalmers University of Technology, 41296 Gothenburg, Sweden; anna.martinelli@chalmers.se (A.M.); vavra@chalmers.se (S.V.)

[*] Correspondence: creci@chalmers.se

**Abstract:** The isomorphous substitution of Si with metals other than Al in zeotype frameworks allows for tuning the acidity of the zeotype and, therefore, to tailor the catalyst's properties as a function of the desired catalytic reaction. In this study, B, Ga, and Ti are incorporated in the MFI framework of silicalite samples and the following series of increasing acidity is observed: Ti-silicalite < B-silicalite < Ga-silicalite. It is also observed that the lower the acidity of the sample, the easier the methanol desorption from the zeotype surface. In the target reaction, namely the direct conversion of methane to methanol, methanol extraction is affected by the zeotype acidity. Therefore, the results shown in this study contribute to a more enriched knowledge of this reaction.

**Keywords:** zeotype; MFI; infrared; methanol

## 1. Introduction

To convert methane to methanol by means of a one-step process is not facile. However, the need for a more efficient and sustainable route towards methanol production, compared to syngas based techniques, has resulted in a tremendous amount of studies in the literature in the last few decades [1–13]. Several technologies have recently been investigated for the direct conversion of methane to methanol (DCMM) [14]. Among these, metal-functionalized zeolites used as heterogeneous catalysts represent one of the most promising since zeolites can host sites similar to the active sites of some enzymes, particularly the methane monooxygenases (MMO) that are known to selectively convert methane to methanol [15]. For this reason, zeolites have been intensively studied in the last few decades for the DCMM. When zeolites are used as catalysts, the DCMM needs to be carried out in a multi-step process for appreciable values of both activity and selectivity. These steps usually consist of the formation of the active site, the activation of methane and the extraction of methanol [1–3]. Several factors have been shown to make the extraction of methanol one of the most difficult and challenging steps in the DCMM. Once methane is activated, it has been shown that most of the methanol produced tend to remain adsorbed on the catalyst surface [4]. The use of solvents, e.g., $H_2O$, during $CH_3OH$ extraction has been shown, both experimentally [2,5] and theoretically [6,16,17], to promote $CH_3OH$ desorption and therefore to increase the production of the desired product. It is also important to avoid methanol over-oxidation, ultimately to $CO_2$, and the formation of unwanted products, e.g., dimethyl ether (DME) [4,5,18,19].

It is therefore crucial to tailor the catalyst's material properties to facilitate methanol extraction, and to study the affinity between methanol and the catalyst surface in order to gain knowledge about the last step in the DCMM. It has been shown that the key reaction intermediates in the DCMM are methoxy groups [3], and that they tend to be strongly adsorbed on the Brønsted acid sites (BASs) of the zeolite [2,3,20,21]. Thus, to decrease

the acidity strength of the BASs would mean to decrease the adsorption strength of these sites towards methanol, or its precursors. Furthermore, it should also be mentioned that a higher Brønsted acidity of the zeolite has been shown to promote the conversion of $CH_3OH$ to DME [22]. Therefore, to decrease the acidity of the BASs would also mean to avoid the formation of undesired products in the DCMM. A way to influence the strength of the Brønsted acidity in zeolite is to incorporate metals other than Al in the zeolitic framework [22–26]. Indeed, the isomorphous substitution of Si with metals of various electronegativity and ionic radii results in –OH groups of the BASs with a stronger or weaker covalent character, and therefore with varying Brønsted acidity [25]. For example, it has been shown that the acidic properties of MFI with Ga and B incorporated in the framework structure change in the sequence Ga » B [27–29]. Henceforth, these Al-free materials showing typical zeolitic structure will be referred to as zeotypes.

The aim of this work is to enrich former studies regarding $CH_3OH$ desorption from zeotypes with varying tuned acidity [26,30]. In those studies, Fe and/or Al were incorporated in the MFI framework of silicalite samples and a pure silicalite sample was prepared as reference. The following series of increasing acidity was outlined: 0 = pure silicalite < Fe-silicalite < Al-silicalite. Furthermore, it was also shown that stronger BASs hinder $CH_3OH$ desorption. In order to further investigate the relationship between Brønsted acidity and methanol adsorption, here B and Ga are incorporated in the MFI framework of silicalite samples and a Ti-silicalite sample is prepared as reference (samples labeled as B-S, Ga-S, and Ti-S). Indeed, no Brønsted acidity is expected in the latter sample since the oxidation state of Ti is +4 and, therefore, no cation is needed to maintain the electroneutrality of the framework [31]. Electron microscopy, X-ray diffraction, and nitrogen sorption are used to characterize the morphology and the structure of the prepared crystallites. The activation of the zeotype samples from the $NH_4^+$- to their $H^+$-form, as well as the final Brønsted acidity, is examined in situ with diffuse reflectance Fourier transform spectroscopy (DRIFTS). Moreover, the evolution of surface species during methanol temperature programmed desorption ($CH_3OH$-TPD) is monitored with DRIFTS. Albeit aware that $CH_3OH$-TPD does not fully reproduce the last step in the DCMM reaction cycle, this experiment is performed to simulate the extraction of $CH_3OH$ in the DCMM.

## 2. Results and Discussion

### 2.1. Materials Characterization

The SEM images show high crystallinity of the zeotype samples (Figure 1 and Figure S1 in the Supplementary Information). However, crystallites with different morphologies are observed depending on the metal added during the synthesis. B-S shows individual crystals with homogeneous geometry, and a uniform particle size distribution with dimensions ranging between 10 and 20 μm. Ga-S, instead, shows interconnected crystals with irregular shapes and a wider particle size distribution. Lastly, Ti-S shows overall more brittle crystals, with features of both B-S and Ga-S. The SEM images of the samples in the $H^+$-form (Figure S1d–f in the Supplementary Information) show that irregular structures are formed on the external facets of the zeotypes crystals during calcination. These structures might be due to the migration of B, Ga and Ti from the T-sites due to instability in the MFI framework. However, since we can conclude from the DRIFTS experiments that metal is still in the MFI framework of the silicalite samples in the $H^+$-form, these structures might also consist of SDA or other synthesis residues which were not eliminated during calcination. Furthermore, the overall morphology of the crystals is maintained after calcination, indicating the thermal stability of the zeotype samples.

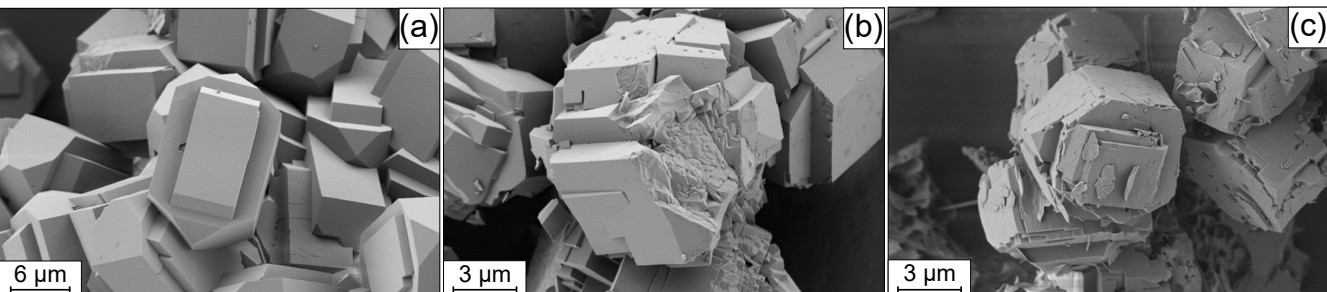

**Figure 1.** SEM images of the as synthesized (**a**) B-, (**b**) Ga- and (**c**) Ti-silicalite samples.

The MFI type of framework structure of all samples is confirmed by XRD [32]. In Figure 2a, the X-ray diffractograms of the as synthesized samples are shown, together with the Miller indices of the crystal planes associated with the diffraction peaks with the highest intensities (indices taken from the Database of Zeolite Structure [33]). No peaks other than those of the MFI framework can be observed, indicating that no additional crystalline phases with long range order are formed. Likewise, no significant peak shifts due to the metal incorporation in the MFI framework can be observed, suggesting that the amount of metal atoms used in the synthesis gel is likely too low for appreciable peak shifts. Furthermore, it is likely that not all metal used in the synthesis gel has isomorphically substituted Si in the MFI framework structure [34], making the peak shift in the X-ray diffractograms even less appreciable. The X-ray diffractograms of the samples in the $H^+$-form (Figure S2 in the Supplementary Information) show a variation of some peak intensities and the disappearance of the diffraction peaks around $2\theta = 12°$ associated with the elimination of the SDA [33]. However, neither extra peaks nor shifts of peak positions can be observed in the diffractograms of the calcined samples, which further confirms the thermal stability of the MFI framework.

The values of the specific BET surface area and micropore volume (see Table 1) are all reasonable for the MFI type of framework structure [35]. That Ti-S shows the lowest values might be due to the fact that the crystal ionic radius of $Ti^{4+}$ (75 pm) is much larger than the one of $Si^{4+}$ (54 pm) [36]. However, $Ga^{3+}$ has a crystal ionic radius of 76 pm, and Ga-S shows higher values of both BET surface area and micropore volume than Ti-S. Therefore, another explanation might be the presence of Ti species in extra-framework position that either have not been successfully incorporated in the framework during synthesis, or migrated from the framework during calcination. Indeed, it is well accepted in the literature that high Ti loadings in the zeotype framework might result in extra-framework $TiO_x$ clusters [37]. However, this Ti-S shows that the typical MFI microporous structure is confirmed both by XRD and the $N_2$ sorption isotherms.

**Table 1.** Specific BET surface area (SSA) and micropore volume of the B-, Ga- and Ti-silicalite samples in the $H^+$-form.

| Sample | SSA [m$^2$ g$^{-1}$] | Micropore v. [$10^{-3}$ cm$^3$ g$^{-1}$] |
|---|---|---|
| B-S | 405 | 78 |
| Ga-S | 335 | 103 |
| Ti-S | 265 | 25 |

The $N_2$ sorption isotherms of all samples (Figure 2b) are of Type IV of the IUPAC classification [38] and therefore further confirm the microporous structure of the zeotype samples. However, B-S shows Type H1 of hysteresis, Ga-S shows Type H4 and Ti-S a combination of both. Type H1 of hysteresis indicates the presence of well-defined channels with uniform sizes and shapes [38,39]. A previous study [26] showed that a metal-free pure silicalite sample exhibits Type IV isotherms and Type H1 hysteresis. Therefore, it is reasonable to assume that B-S presents a more similar microporous structure to the pure MFI framework. Type H4 of hysteresis, instead, indicates materials containing

both micropores and mesopores and is often found in zeolitic materials with aggregated crystals [38,40]. Therefore, the microporous structure of Ga-S seems to present a less uniform geometry. The interpretations from the nitrogen sorption isotherms are indeed supported by the SEM images (Figure 1 and Figure S1 in the Supplementary Information), where it was observed that B-S is characterized by independent single crystals, whilst Ga-S shows agglomerates of crystals that might cause the formation of mesopores. In this perspective, the isotherms of Ti-S suggest the presence of both a well-defined porous structure and mesopores. Indeed, the SEM images of Ti-S show both well-separated crystals and agglomerates of crystals.

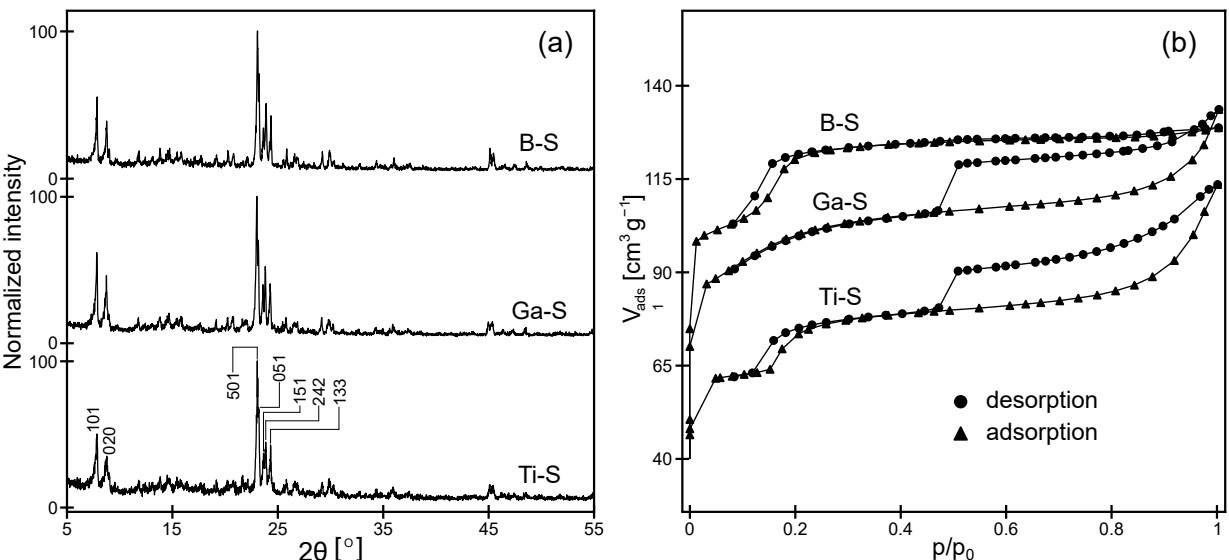

**Figure 2.** (**a**) X-ray diffractograms of the as synthesized B-, Ga- and Ti- silicalite samples; (**b**) nitrogen adsorption and desorption isotherms of the B-, Ga- and Ti- silicalite samples in the $H^+$-form.

In order to investigate the acidity of the samples, in situ IR spectra were collected during the activation of the $NH_4^+$-form of all samples and after calcination at room temperature. As an example, the IR spectral series collected during calcination of the $NH_4^+$-form of Ga-S is shown in Figure 3a (the corresponding IR spectral series of B-S and Ti-S are shown in Figure S3a and b, respectively, in the Supplementary Information). The positive absorption peak in the O–H stretching region at 3721 $cm^{-1}$ is assigned to terminal isolated silanol groups (Si–OH) and is typical of many silica-based materials. It does not significantly change in intensity as the temperature of the DRIFTS cell is increased, meaning that these silanol groups are not involved in the zeotype activation process. That the positive absorption peak in the O–H stretching region at 3608 $cm^{-1}$ increases in intensity as the temperature of the DRIFTS cell is increased, indicates, instead, that this peak is related to –OH species whose amount increases during the activation process of the zeotype. Since the broad negative absorption mode in the N–H stretching region around 3300 $cm^{-1}$, associated with the decomposition of $NH_4^+$ during the activation process, evolves together with the peak at 3608 $cm^{-1}$, the absorption peak at 3608 $cm^{-1}$ is assigned to O–H stretching vibrations of the –OH groups of the Brønsted acid sites generated by Ga ($\equiv Si-O(H)-Ga\equiv$). B-S shows similar behavior as Ga-S (Figure S3a in the Supplementary Information), but the peak assigned to the BASs generated by B is shifted to higher wavenumbers. Ti-S shows only one sharp absorption peak in the O–H stretching vibration region at 3735 $cm^{-1}$ (Figure S3b in the Supplementary Information), assigned to the O–H stretching of Si–OH. Indeed, Ti-S is not expected to present any Brønsted acidity since the oxidation state of Ti is +4 and therefore the electroneutrality of the framework is maintained when $Ti^{4+}$ substitutes $Si^{4+}$ [31].

The IR spectra recorded at room temperature after the activation of the $NH_4^+$-form of the zeotype samples are shown in Figure 3b. The absorption peaks at 3745 $cm^{-1}$ for Ti-S

and 3728 cm$^{-1}$ for B-S and Ga-S are assigned to the O–H stretching vibration of Si–OH. Therefore, these hydroxyl groups correspond to the –OH whose vibration modes result in the absorption peaks at 3735 cm$^{-1}$ in Figure S3b, 3723 cm$^{-1}$ in Figure S3a and 3721 cm$^{-1}$ in Figure 3a, respectively. The absorption peaks at 3616 cm$^{-1}$ for Ga-S and 3709 cm$^{-1}$ for B-S in Figure 3b are assigned to the O–H stretching vibration of the –OH groups belonging to the Ga- and B-BAS. Therefore, these hydroxyl groups correspond to the –OH whose vibration modes results in the absorption peaks at 3608 cm$^{-1}$ in Figure 3a and 3695 cm$^{-1}$ in Figure S3a, respectively. It should be noted that these temperature-dependent shifts of absorption peaks assigned to the vibrational mode of the same species are consistent for all samples and are commonly observed in IR spectroscopy [41].

The incorporation in the MFI framework of more electronegative elements leads to more covalent –OH groups belonging to the BASs [25]. As previously observed in literature [23–25] and in a former study by the authors [26], this is translated in the IR spectra in a blueshift of the peak assigned to the O–H stretching of the BASs, and thus the following series of increasing Brønsted acidity can be outlined: 0 = Ti-S < B-S < Ga-S. With respect to the intensity of the absorption peaks at 3728 cm$^{-1}$ of B-S and Ga-S, it is noteworthy that the intensity of the peak at 3709 cm$^{-1}$ is weaker than the intensity of the peak at 3616 cm$^{-1}$. This suggests that, with respect to the isolated terminal silanol groups of each sample, there are more Ga-BASs than B-BASs. Furthermore, it should also be observed that the absorption peak assigned to Si–OH is centered at higher wavenumbers, sharper and more pronounced in Ti-S than in B-S and Ga-S, indicating that the terminal silanol groups are more isolated in Ti-S. This might be due to the fact that Ti-S shows no Brønsted acidity and therefore the terminal silanol groups have, for example, fewer possibilities to interact with –OH groups of other nature.

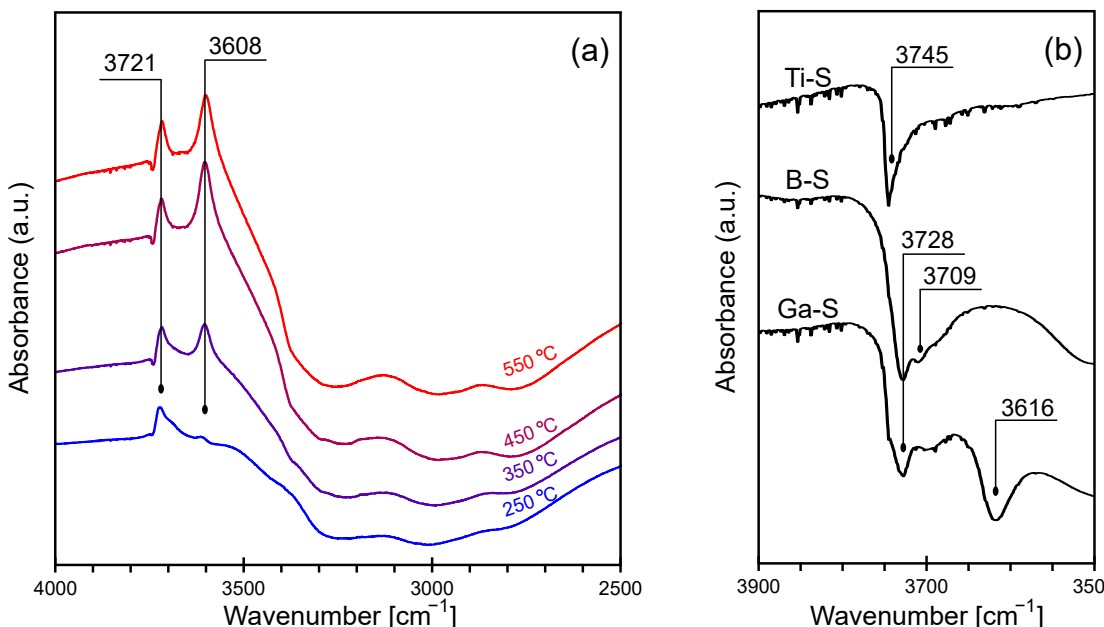

**Figure 3.** (**a**) Background subtracted IR spectra of the Ga-silicalite sample in the NH$_4^+$-form recorded during calcination in 10% O$_2$/Ar. The background was taken at 200 °C in 10% O$_2$/Ar; (**b**) IR spectra recorded at room temperature in Ar of the B-, Ga- and Ti- silicalite samples in the H$^+$-form.

### 2.2. Temperature Programmed Desorption of Methanol

In Figure S4 in the Supplementary Information, the complete spectral series of Ga-S collected during CH$_3$OH-TPD is shown. It is clear that the C–H and the O–H stretching regions are the most relevant in order to investigate the evolution of the surface species during the experiment. Therefore, particular attention will be given to these two regions for the spectral series of the other samples. In Figure 4a, the C–H stretching region of

the IR spectra of all samples recorded at 50, 300 and 550 °C after $CH_3OH$ adsorption is shown. For clarity, the IR spectra related to the intermediate temperature steps are not shown as they would not give additional information to the study. Since Si and Ti have the same oxidation state in the MFI framework, the isormorphous substitution of $Si^{4+}$ with $Ti^{4+}$ does not form Brønsted acidity as revealed in Figure 3b and Figure S3b, and therefore Ti-S is not expected to have strong adsorption sites. It is thus reasonable that the spectra of Ti-S show absorption peaks with overall lower intensity, indicating lower adsorption strength towards $CH_3OH$ than B-S and Ga-S. Furthermore, the general scenario of Figure 4a accords with former studies performed on pure, Fe- and Al-silicalite samples [30]. Indeed, the methyl groups belonging to the adsorbed methanol or the methoxy species have different types of symmetries depending on the structure of the adsorption sites. Different $-CH_3$ symmetries lead to different numbers of non-degenerate C–H stretching vibrations, and therefore different numbers of relative absorption peaks can be observed in the IR spectra.

Liquid methanol shows methyl groups with $C_s$ type of symmetry. Indeed, three absorption peaks at 2980, 2946 and 2834 $cm^{-1}$ are observed in the C–H stretching region of the IR spectra of liquid methanol, which are assigned to the out-of-plane antisymmetric, the in-plane antisymmetric and the symmetric C–H stretching vibration, respectively [42]. At 50 °C, methanol is hydrogen bonded to the zeotype surface and the methyl groups have therefore a $C_s$ type of symmetry as well. The absorption peaks in Figure 4a at 2998 and 2954, and 2852 $cm^{-1}$ are thus assigned to the two antisymmetric and the symmetric C–H stretching vibrations of methanol hydrogen bonded to the zeotype surface. The blueshift of the absorption peaks in the spectra of adsorbed $CH_3OH$ in Figure 4a compared to liquid $CH_3OH$ [42] is due to the interaction of the molecule with the zeotype surface and has previously been observed [20,43]. Ti-S shows additional absorption peaks around 2930 and 2830 $cm^{-1}$. With regard to the first peak, a pronounced shoulder at 2930 $cm^{-1}$ has previously been assigned to an overtone or a combination of the bending modes of the methoxy species belonging to methanol adsorbed at room temperature on pure silicalite [44]. To the authors' knowledge, no assignment has previously been done in the literature to absorption peaks around 2830 $cm^{-1}$ for silicalite samples after $CH_3OH$ adsorption. However, this peak markedly decreases in intensity as the temperature of the DRIFTS cell is increased, and ultimately disappears at 200 °C. It is thus reasonable to assume that the adsorbed species whose vibrations result in the peak around 2830 $cm^{-1}$, are not strongly bound to the surface of Ti-S.

As the temperature of the DRIFTS cell is increased, the physisorbed methanol desorbs, leaving only the methoxy species that are more strongly bound to the zeotype adsorption sites. Assuming a $C_{3v}$ type of symmetry of the adsorbed methoxy groups, two peaks are expected for each type of adsorption site (two degenerate for the antisymmetric and one for the symmetric C–H stretching vibrations). The absorption peaks at 2956 and 2855 $cm^{-1}$ have previously been assigned to the antisymmetric and symmetric C–H stretching vibration of $\equiv Si-O(CH_3)$ species, respectively [20,45]. Absorption peaks around 2980 $cm^{-1}$ were previously observed after $CH_3OH$ adsorption on Al-containing ZSM-5 zeolites. It is widely accepted that the latter absorption peak is assigned to the antisymmetric C–H vibration of methoxy groups strongly bound on BASs generated by Al after $CH_3OH$ adsorption [13,20,46,47]. Furthermore, in a former study [30], an absorption peak at 2978 $cm^{-1}$ was assigned to the antisymmetric C–H vibration of methoxy groups adsorbed on BASs generated by Fe after $CH_4$ exposure. Therefore, it seems that the type of metal which generates the BASs does not significantly influence the position of the peak assigned to the C–H stretching of the methoxy groups adsorbed on the BASs.

Based on these considerations, we assign the absorption peak at 2978 $cm^{-1}$ in Figure 4a to the antisymmetric C–H stretching vibrations of methoxy groups adsorbed on BASs generated by Ga ($\equiv Si-O(CH_3)-Ga\equiv$). This assignment represents a novelty in the literature to the authors' knowledge. It is likely that the absorption peak relative to the symmetric stretching is not observed or is covered by the other peaks, since antisymmetric vibrations

are usually more IR active than the relative symmetric ones, and the peak at 2978 cm$^{-1}$ shows low intensity compared to the peaks at 2956 and 2855 cm$^{-1}$. This assignment is further supported by Figure 4b, where the O–H and C–H stretching regions of the CH$_3$OH-TPD IR spectral series of Ga-S is shown wih smaller temperature-steps than in Figure 4a. As soon as the positive peak at 2978 cm$^{-1}$ forms at around 250 °C, a negative peak at 3618 cm$^{-1}$ forms as well, and these two peaks evolve together as the temperature of the DRIFTS cell is increased. Since the absorption peak at 3618 cm$^{-1}$ was assigned to the O–H stretching vibration of Ga-BASs (Figure 3b), Figure 4b clearly shows that methoxy groups become IR visible at the expenses of –OH groups belonging to Ga-BASs, confirming our assignment of the absorption peak at 2978 cm$^{-1}$. It is noteworthy that B-S shows no peaks other than the ones assigned to the C–H stretching vibration of $\equiv$Si$-$O(CH$_3$), suggesting that B-BASs are not strong adsorption sites towards CH$_3$OH.

The IR spectrum of Ga-S is the only one that shows absorption peaks at 550 °C, indicating some methoxy groups are still adsorbed on the zeotype surface. B-S and Ti-S show no absorption peaks suggesting that most methoxy species are desorbed in the last step of the TPD experiments. Lastly, it should be mentioned that a quantitative analysis of the intensities of the absorption peaks at 2956 and 2855 cm$^{-1}$, and the absorption peak at 2978 cm$^{-1}$ could lead to misunderstandings of the roles of the adsorption sites. Indeed, they are assigned to vibrations of species adsorbed on sites that occur in different quantities in the zeotype samples (Si:M = 50, M = B, Ga or Ti). Moreover, the absorption peak at 2978 cm$^{-1}$ has higher intensity in the spectrum recorded at 550 °C than in the spectrum recorded at 300 °C, with respect to the peak at 2956 cm$^{-1}$. This indicates that, at higher temperatures, more methoxy species are adsorbed on Ga-BASs than on silanol groups, with regard to all ad-species, confirming the strong adsorption of methanol on Ga-BASs.

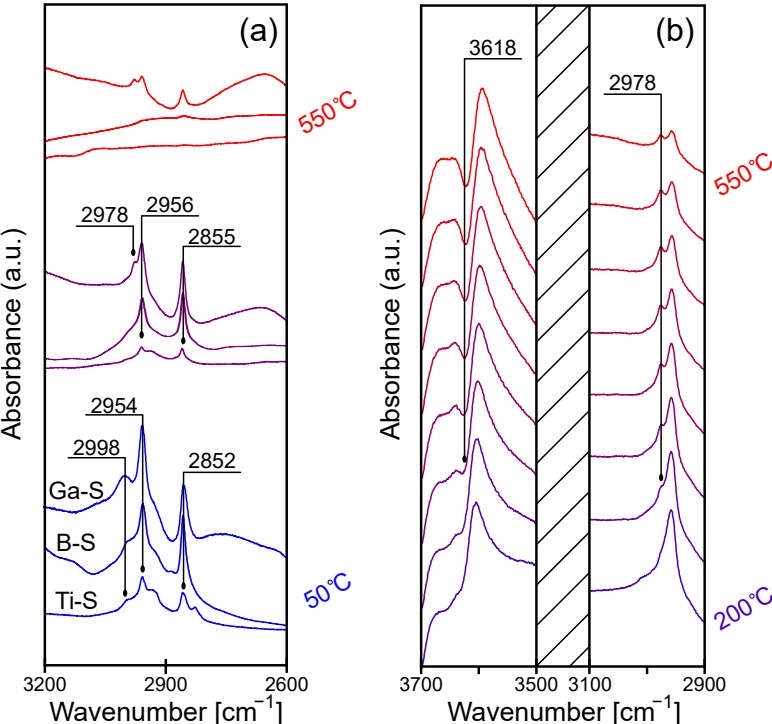

**Figure 4.** (**a**) Background subtracted CH$_3$OH-TPD IR spectra recorded in Ar at 50, 300 and 550 °C of the Ti-, Ga- and B-silicalite samples in the H$^+$-form. The background spectra were taken in Ar at room temperature after the 10% O$_2$/Ar pretreatment and before CH$_3$OH adsorption; (**b**) background subtracted CH$_3$OH-TPD IR spectral series recorded in Ar from 200 to 550 °C (steps of 50 °C) of the Ga-silicalite sample in the H$^+$-form. The background spectrum was taken in Ar at room temperature after the 10% O$_2$/Ar pretreatment and before CH$_3$OH adsorption.

### 2.3. Further Considerations on the Zeotypes' Acidity and Their Applications

The acidity in zeolites is a crucial parameter for countless chemical reactions. Zeolites are, for example, widely used in many acid-catalyzed reactions: fluid catalytic cracking and hydrocracking, as well as isomerization and alkylation reactions, to mention some of the most common [48]. In addition, several studies have shown that acidity plays an important role in metal-catalyzed reactions as well, where different metals are hosted in various positions in the zeolite structure. Complete methane oxidation using Pd-ZSM-5 [49], direct conversion of methane to methanol using Fe-ZSM-5 [18] and $NH_3$ selective catalytic reduction using Cu-SSZ-13 [50] are some examples of reactions where the metals are the active species, but the acidity of the zeolite significantly influences the activity and the selectivity of the catalysts. Therefore, the results shown in this work not only can help in the fundamental understanding of the DCMM, but also pinpoint important catalyst design principles for other catalytic reactions.

### 3. Materials and Methods

Samples of silicalite with B (B-S), Ga (Ga-S) and Ti (Ti-S) with the MFI type of framework structure were prepared according to the method by Szotak et al. [51]. The chemicals used for the synthesis were: tetraethyl orthosilicate (TEOS, reagent grade, 98%, Sigma Aldrich, MO 63103, USA) as silica source, boric acid (Fluka Chemicals, Fisher Scientific, Gothenburg, Sweden), gallium (III) nitrate solution (Ga 9–10 wt.%, Alfa Aesar, Thermo Fisher, Kandel, Germany) and titanium (IV) ethoxide (Aldrich, 3050 Spruce Street, Saint Louis, MO, USA) as metal precursors, and tetrapropylammonium hydroxide (TPAOH, 1.0 M Lsg. in water, Sigma-Aldrich, CH-9471 Buchs (SG), Switzerland) as structure directing agent (SDA). Oxalic acid (98%, Sigma-Aldrich) and sodium hydroxide (pellets, 98%, Alfa Aesar) were used to adjust the pH value. The calculated molar ratio Si:M:TPA$^+$:H$_2$O = 1:0.02:0.300:24.86 was kept constant in all syntheses (M = B, Ga or Ti).

TEOS was hydrolyzed for 24 h at 60 °C in water and oxalic acid. The metal precursors were dissolved in water for 5 h and added to the TEOS containing solution. After 4 h stirring, a 20 wt.% solution of TPAOH was added drop-wise under vigorous stirring. The pH of the solution was then adjusted with a 1M NaOH solution to the basic conditions necessary to promote crystallization (pH = 11). The gels were heated in unstirred autoclaves (0.2 l, Parr Instrument Company, Moline, IL 61265-1770, USA) for 5 days at 170 °C. The powder samples were obtained by washing, filtration and freeze-drying of the crystallization products. The as synthesized samples were calcined at 500 °C for 5 h (5 °C min$^{-1}$) in air in order to remove the SDA and to obtain the Na$^+$-form of the zeotypes. The Na$^+$-zeotype samples were then ion-exchanged twice at 60 °C for 24 h with ammonium nitrate (EMPLURA®, Sigma-Aldrich, Darmstadt, Germany), followed by washing, filtration and freeze-drying overnight. The H$^+$-form of the zeotype was obtained by calcination at 500 °C for 1 h (10 °C min$^{-1}$) in air.

Images of the zeotype crystallites were taken using a Zeiss Ultra 55 FEG scanning electron microscope (Jena, Germany) on samples that were coated with palladium using an EMITECH K550x sputter coater (France). The X-ray diffractograms of the SDA-containing samples and the samples in the H$^+$-form were recorded with a Bruker XRD D8 advanced instrument (Billerica, MA, USA) with monochromatic CuK$\alpha$ radiation scanning from 5 to 55°. Nitrogen sorption was performed with a Micrometrics Tristar 3000 instrument (Norcross GA, USA) at −196 °C after degassing in N$_2$ at 220 °C for 5 h.

The in situ infrared spectra were recorded using a VERTEX 70 spectrometer (Bruker) in the diffuse reflectance mode equipped with a liquid nitrogen cooled mercury cadmium telluride (MCT) detector (bandwidth 600–12,000 cm$^{-1}$), a Praying Mantis™ accessory and a high-temperature stainless steel reaction chamber (Harrick Scientific Products, Inc., Ossining, NY, USA). All spectra were measured between 800 and 4000 cm$^{-1}$ with a resolution of 1 cm$^{-1}$. The samples were sieved and the fraction between 40 and 80 μm was used for analysis. For each measurement, the sample bed was supported by a KBr bed to facilitate the use of low amounts of sample.

In order to investigate the acidity of the zeotype samples, IR spectra were collected during the activation of the samples from the $NH_4^+$- to the $H^+$-form (10% $O_2$/Ar, from 200 to 500 °C, heating rate of 10 °C min$^{-1}$) and after calcination. In order to investigate the affinity between $CH_3OH$ and the surface of the zeotype samples, a few droplets of $CH_3OH$ (99.8% Sigma-Aldrich) were added to each sample in the $H^+$-form and the evolution of the surface species was followed by in situ DRIFTS while increasing the cell temperature. The TPD experiments consisted of step-wise temperature increases from 30 to 550 °C in Ar, and each spectrum was recorded when the target temperature was stable. Before the $CH_3OH$-TPD experiments, all samples were pretreated in 10% $O_2$/Ar for 1 h at 500 °C (starting from room temperature with a heating rate of 10 °C min$^{-1}$) to remove any possible contamination from the surface. The background spectra were taken in Ar at room temperature after the pretreatment and before $CH_3OH$ exposure.

## 4. Conclusions

The results from the morphological and structural characterization indicate that all samples show the MFI type of framework structure. However, B-S shows individual crystallites, whilst Ga-S shows agglomerates that result in a structure consisting of micro- and mesopores. The crystallites of Ti-S show features of both B-S and Ga-S. The calcination of all samples leads to the formation of structures on the external facets of the crystallites, likely due to synthesis residuals. However, the MFI framework is maintained after calcination, indicating the thermal stability of the zeotype samples. The IR spectra of the $H^+$-form of all samples allow for outlining the following series of increasing acidity: 0 = Ti-silicalite < B-silicalite < Ga-silicalite. Indeed, a shift of the absorption peak assigned to the O–H stretching of the Brønsted acid sites is observed depending on the metal incorporated in the MFI framework. This assignment is further confirmed by the simultaneous decomposition of the -$NH_4^+$ and the formation of the –OH groups of the BASs during the activation of the zeotype samples from the $NH_4^+$- to the $H^+$ forms.

The $CH_3OH$-TPD experiments show that methanol is overall more weakly bound to the zeotype surface when the zeotype acidity is lower. Furthermore, methoxy groups strongly bound to Brønsted acid sites are observed only when Ga is incorporated in the MFI framework. In particular, an absorption peak at 2978 cm$^{-1}$ is observed for Ga-S during the $CH_3OH$-TPD experiments at high temperature. This absorption band has not been observed before to the authors' knowledge and is here assigned to the C–H stretching of methoxy groups adsorbed on Ga-BASs. Ga-S is also the only sample showing ad-methoxy groups at high temperatures, indicating the high adsorption strength of Ga-S towards $CH_3OH$. In summary, this work confirms previous studies by the authors, where IR and $CH_3OH$-TPD experiments with pure, Fe- and Al-silicalite samples were performed. The lower the acidity of the zeotype, the easier it is to desorb $CH_3OH$ from the zeotype surface. Since $CH_3OH$ extraction is one of the key steps in the direct conversion of methane to methanol, these results clearly indicate that the acidity of the zeotype is a determining parameter for this reaction.

**Supplementary Materials:** The following are available online at https://www.mdpi.com/2073-4344/11/1/97/s1: Figure S1: SEM images of the B-, Ga- and Ti-silicalite samples in the as synthesized and $H^+$-form. Figure S2: X-ray diffractograms of the B-, Ga- and Ti- silicalite samples in the $H^+$-form. Figure S3: IR spectra of (**a**) the B- and (**b**) Ti-silicalite samples in the $NH_4^+$-form recorded during calcination in 10% $O_2$/Ar. Figure S4: Complete $CH_3OH$-TPD IR spectral series recorded in Ar from 25 to 550 °C (steps of 25 °C) of the Ga-silicalite sample in the $H^+$-form.

**Author Contributions:** Conceptualization, S.C. and M.S.; methodology, S.C. and M.S.; software, S.C.; validation, S.C. and M.S.; formal analysis, S.C.; investigation, S.C. and S.V.; resources, M.S. and A.M.; data curation, S.C.; writing—original draft preparation, S.C.; writing—review and editing, S.C., A.M., S.V., P.-A.C. and M.S.; visualization, S.C.; supervision, A.M., P.-A.C. and M.S.; project administration, M.S.; funding acquisition, M.S. All authors have read and agreed to the published version of the manuscript.

**Funding:** This work was performed in part at the Chalmers Materials Analysis Laboratory, CMAL. The work was financially supported by the Swedish Research Council through the Röntgen-Ångström collaborations "Time-resolved in situ methods for design of catalytic sites within sustainable chemistry" (No. 349-2013-567), the Knut and Alice Wallenberg foundation "Atomistic design of catalysts" (No. 2015.0058), and the Competence Centre for Catalysis, which is hosted by Chalmers University of Technology and financially supported by the Swedish Energy Agency (No. 22490-4) and the member companies AB Volvo, ECAPS AB, Johnson Matthey AB, Preem AB, Scania CV AB, Umicore Denmark ApS and Volvo Car Corporation AB. A.M. and S.V. acknowledge funding from the Swedish Foundation for Strategic Research (SSF, FFL-6 program, Grant No. FFL15-0092) and from the Knut and Alice Wallenberg Foundation (Academy Fellows program, Grant No. 2016-0220).

**Data Availability Statement:** The data presented in this study are available within the article and the supplementary materials.

**Conflicts of Interest:** The authors declare no conflict of interest. The funders had no role in the design of the study; in the collection, analyses, or interpretation of data; in the writing of the manuscript, or in the decision to publish the results.

## Abbreviations

The following abbreviations are used in this manuscript:

| | |
|---|---|
| XRD | X-ray diffraction |
| SEM | Scanning electron microscopy |
| DRIFTS | Diffuse reflectance Fourier transform spectroscopy |
| B-S | B-silicalite |
| Ga-S | Ga-silicalite |
| Ti-S | Ti-silicalite |

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
