# Peer review of "Acidity as Descriptor for Methanol Desorption in B-, Ga- and Ti-MFI Zeotypes"

_catalysts, doi:10.3390/catal11010097_

Round 1
Reviewer 1 Report
The manuscript is within the scopes of the SI. I recommend the acceptation of the manuscript after a minor revision. Before publication the authors must clarify the following aspects:
Since the acidity of titanium modified zeolite and pure silicalite (MFI) is zero, can you clarify the role of titanium in the adsorption of methanol? Has the Brönsted acidity of the different samples been measured, e.g. through TPD of ammonia?
There are some errors such as.:
Row 32: BASs must be in bracket.
Row 117: the sentence “That the positive absorption peak in the O–H stretching region at 3608 cm-1 increases in intensity as the temperature of the DRIFTS cell is increased, indicates, instead, that this peak is related to –OH species involved in the activation process of the zeotype.” is not clear.
Row 147: “This suggests that, with respect to the isolated terminal silanol groups, there are more Ga-BASs than B-BASs”. What is more? What is it about, for the terminal silanol groups, please explain?
Reviewer 2 Report
The manuscript entitled “Acidity as descriptor for methanol desorption in B-, Ga- and Ti-MFI zeotypes”, by Simone Creci et al. is interesting and the catalysts were characterized by variety of techniques. However, there are pending tasks before the manuscript achieve the required quality. Following please find the main observations:
- The effect of the acidic strength of the protons in the bridged Si(OH)MIII (M = B and Ga) groups was investigated in many works which results should be discussed in the Introduction part. For example, the following works demonstrated that the acidic property of the MFI are changing according to the sequence Ga >> B:
C.T.-W. Chu, C. Chang. J. Phys. Chem., 89 (1985) 1569 ; Strodel et al. Chem. Phys. Lett., 240 (1995) 547; M.S. Stave, J.B. Nicholas. J. Phys. Chem., 99 (1995) 15046; D.J. Parrillo et al. J. Phys. Chem., 99 (1995) 8745.; R. Aiello et al. Comtes Rendus Chime 8 (2005) 321.)
- The catalyst characterization by XRD did not show evidence of the B and Ga incorporation into the MFI framework. Indeed, taking into account that B-S catalyst was prepared using the alkaline media (pH=11) and boric acid as B precursor, the introduction of four B per unit cell into the MFI framework is hardly expected (e.g. see Perego et al. Micropor. Mesopor. Mater., 58 (2003) 213). However, the crystal size should be calculated in order to confirm this. Therefore, there is not “the isomorphous substitution of Si with metals “, which is the expression written in Abstract and Introduction (lines 39-41).
- Lines 228-231. It is written “To conclude” but this information is not conclusion of this work. I should be better to include it in the Introduction part.
